# Tropical Cyclone Exposure in the North Indian Ocean

**Rubaiya Kabir** [1,*] **, Elizabeth A. Ritchie** [2,3] **and Clair Stark** [1]

1    School of Science, University of New South Wales, Canberra, ACT 2600, Australia
2    School of Earth, Atmosphere, and Environment, Monash University, Clayton, VIC 3168, Australia
3    Department of Civil Engineering, Monash University, Clayton, VIC 3168, Australia
*    Correspondence: rubaiya.kabir@student.adfa.edu.au

**Abstract:** The North Indian Ocean is a region with a high coastal population and a low-lying delta, making it a high-risk region for tropical cyclone impacts. A 30-year period from 1989–2018 has been used to examine the TC landfalling exposure in the North Indian Ocean and its changes by considering 30 years of IBTrACs data, ERA5 atmospheric data, and 20 years of TRMM and DAV data. A total of 185 TCs made landfall in the NIO during the 30-year period with the majority of the TCs making landfall during the pre- and post-monsoon seasons. Rainfall associated with landfalling TCs decreased in the last 10 years of analysis (2009–2018) compared to the first 10 years of available data from 1999–2008. During the monsoon, TC activity is relatively lower compared to the post-monsoon periods, even though higher accumulated TC-associated rainfall typically occurs during the monsoon period, particularly along the eastern coastlines of the Arabian Sea and the Bay of Bengal. The TC winds impact most of the Bay of Bengal coastline, including Sri Lanka. The spatial distribution of landfalling TCs changes with the season, with most of the landfalling activity occurring during the pre- and post-monsoon periods. Interestingly, more recent TC activity has shifted to the northeast India and Bangladesh coasts, suggesting that these regions may be more vulnerable to TC impacts in the future.

**Keywords:** North Indian Ocean; tropical cyclones; TC activity; impacts; rainfall; winds; tracks





## 1. Introduction and Background

Tropical cyclones (TCs) are one of the most intense and impactful weather systems in the world. They are extremely important for the weather and climate of many tropical countries [1] since they bring much of the annual rainfall to many tropical regions. However, they can also produce periods of extremely heavy rain, which can result in flooding and landslides, particularly in complex terrain along with high winds. As a consequence, it is important to understand how the behaviour of TCs, such as their frequency and intensity and their physical impacts, will change due to changes in the climate [2]. Hence, understanding the links between climate and TCs is a topic of substantial scientific interest [3], motivated in large part by the societal and economic impact of hurricanes [4–6].

Landfalling TCs can produce several different physical hazards, notably storm surges, high seas, strong winds, heavy rain, flooding, and related mud/landslides [7]. The damage caused by TCs to marine interests both over the open ocean and near coastal habitats has been recognized for hundreds of years [1]. Storm surges, strong winds and flooding can produce considerable loss of life and damage to property every year in the affected regions. Climate change is expected to produce sea level rise, which is likely to increase the risk of coastal damage and inundation from storm surges, while increases in atmospheric moisture will likely increase TC-related rainfall inland [8,9].

The North Indian Ocean (NIO) is a relatively small tropical cyclone basin bounded by Africa to the west, Asia to the north and east, and the equator to the south. It is divided into the Arabian Sea and the Bay of Bengal. The climate is dominated by a monsoonal regime driven by the strong heating contrast between the land and the ocean [10,11]. During the

winter, from October to April, strong north-easterly winds dominate the region. From May to October, the winds reverse direction to prevailing south-westerly winds. Up to 80% of the total annual rainfall can fall in the region during the summer monsoon. The climate variation of sea surface temperatures (SSTs) is an important predictor of rainfall in the monsoon season [12]. The monsoon season divides into three monsoon periods: the pre-monsoon (February–May), monsoon (June–September) [13,14], and post-monsoon (October–January) periods. The pre-monsoon period is mostly devoid of TCs, although many monsoon depressions form in the northern part of the Bay of Bengal during May. However, very few of these monsoon depressions ever intensify into cyclones [15]. Typically, more frequent and stronger TCs develop during the early monsoon and post-monsoon periods compared to the main monsoon period [16,17].

On average, approximately 5.5 TCs develop in the NIO annually [18,19], with four in the Bay of Bengal and one and a half in the Arabian, and this comprises about 7% of the global annual total TCs [20–22]. TCs have affected NIO border communities since the earliest days of settlement [23] and can have substantial impacts on the coastal countries of the Bay of Bengal and the Arabian Sea [24]. The frequency of TCs in the North Indian Ocean basin is not high compared with the western North Pacific basin, in part because the ocean area is small. However, the large amount of land exposure means that a large percentage of the TCs that form in the NIO make landfall in the region (Figure 1), with coastal regions in the Bay of Bengal, including India, Bangladesh, and Myanmar being most affected [25–27] (Figure 1b,c).

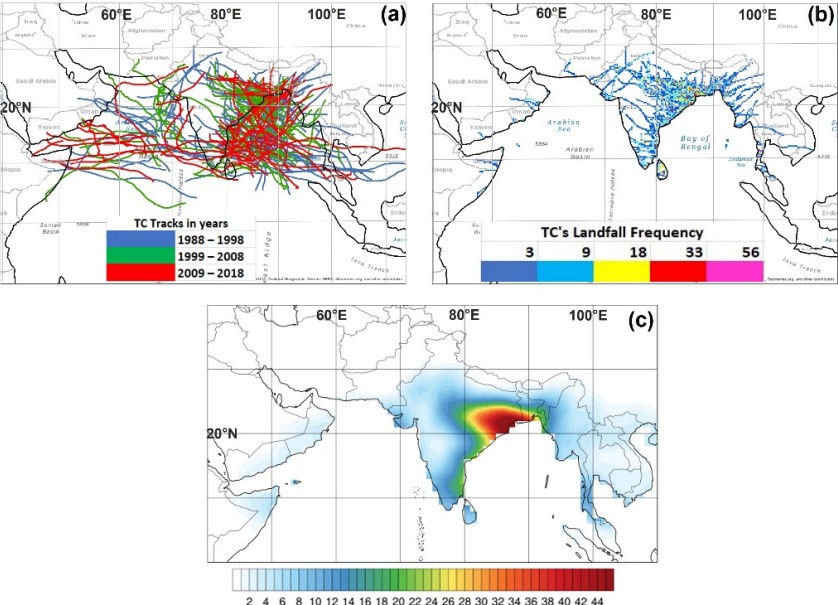

**Figure 1.** North Indian Ocean tropical cyclones that made landfall during 1989–2018: (**a**) full tracks; (**b**) frequency map of landfall locations with inland tracks (grid 25 × 25 km² resolution); and (**c**) track density (0.1° latitude resolution).

TC activity over the NIO is impacted by a number of climate modes. Decadal variability has been identified with a 29-year cycle in tropical cyclone frequency in the NIO [16], which switched in the 1990s to a rising activity period. Activity is also modulated by the El Niño-Southern Oscillation (ENSO) mode, with a reduction in activity in the Bay of Bengal during May and November during the ENSO warm phase [16,28,29]. Activity in the post-monsoon season is enhanced during La Nina, particularly for more intense (>64 kt) cyclones [28]. This is attributed to the existence of low-level cyclonic vorticity, enhanced convection, and high tropical cyclone heat potential in the Bay of Bengal, which provides favorable conditions for TC activity during La Niña [28]. In addition, genesis locations shift

east during La Niña providing a longer path for TCs over warm ocean waters, which likely contributes to the development of higher intense TCs [28].

Similar to the ENSO modulation in the Pacific Ocean, a change in temperature gradients across the Indian Ocean results in changes in the preferred regions of rising and descending moisture and air across the basin. The Indian Ocean Dipole (IOD) is a specific mode of interannual variability of SSTs in the Indian Ocean, which has been found to affect TC annual landfalls in the NIO [30] by modulating the convection and, thus, the large-scale atmospheric circulation pattern. Any relationship between the IOD and ENSO and their potential feedback from each other in the NIO is not well documented [31–35]. However, TC genesis is generally decreased (increased) during the positive (negative) phase of the IOD. In addition, the westerly steering flow over the Bay of Bengal is strengthened (weakened), potentially leading to higher exposure on the eastern coastline of the Bay of Bengal, but by fewer TCs [30]. The NIO TC interannual variability is more closely related to variability in atmospheric circulation patterns than directly to the changing SST patterns. This variability includes higher vertical wind shear, lower horizontal wind shear, and lower mid-level tropospheric moisture, which are associated with periods of lower TC activity in the basin during positive IOD phases compared to negative IOD phases [36,37]. Furthermore, the interannual variability of post-monsoon TCs over the Bay of Bengal is controlled by the interaction between the mid-tropospheric relative humidity and the long-term mean states of absolute vorticity and potential intensity [36,37]. During the post-monsoon period, increased mid-tropospheric moisture favors more frequent TC genesis in negative IOD phases [37].

Finally, NIO TC activity has also been shown to be strongly modulated by intraseasonal variability, including the Madden–Julian Oscillation (MJO) and the boreal summer intraseasonal oscillation (BSIO) [38,39]. This is, in part, because the main NIO activity occurs during the transitional months when the MJO and BSIO are more active rather than during the peak Northern Hemisphere TC period.

There are very few studies that investigate the TC rainfall contribution over the entire NIO basin. However, TC frequency and associated rainfall have been intensively investigated in the Bay of Bengal [40–42]. More generally, across the NIO, landfalling TCs have been associated with heavy rain and associated flooding in the coastal regions of India, Bangladesh, Pakistan, and Myanmar, affecting local agriculture, livestock, houses, and people.

Climate projection studies suggest that in South Asia, there may be an enhancement of summer monsoon precipitation and increased rainfall extremes associated with landfalling TCs on the coastal areas of the Bay of Bengal [17,43,44]. Recent modelling and attribution studies [45–47] suggest that a TC storm surge could reach more than 7 m in height in some regions of the NIO, especially towards the end of the 21st century. Tropical cyclone genesis is also projected to decrease in the Bay of Bengal (22–43%) and increase in the Arabian Sea (30–64%) [48–51], with much of the change occurring in the pre-monsoon season due to changes in vertical ascent supporting convection. However, it is important to note both the large uncertainty in the ranges and that the large percentages apply to very small TC numbers. In contrast, a study using medium resolution ensemble experiments from the NCAR Community Earth System Model (CESM) suggested that tropical depressions in the NIO may increase in the second half of the 21st century [52], highlighting the highly uncertain nature of this region.

Furthermore, also affecting future TC activity in this region are future changes to the monsoon circulation. Projections suggest there may be a significant increase in mean monsoon precipitation of 8% and a possible extension of the monsoon period [53,54]. In addition, an increase in precipitable water of 12–16% is projected over major parts of India. A maximum increase of about 20–24% is found over the Arabian Peninsula, adjoining regions of Pakistan, northwest India, and Nepal [53]. Although the projected summer monsoon circulation appears to weaken, the projected anomalous flow over the Bay of Bengal and Arabian Sea will support oceanic moisture convergence towards the southern

parts of India and Sri Lanka [55]. Variation in environmental moisture is important in modulating TC activity.

The above-mentioned studies have investigated TC activity over the NIO, including projections of how that activity will change in the future. However, there is no comprehensive study of how TCs impact countries after making landfall. Combining the highly exposed population with landfalling TCs (e.g., [44]) can result in extensive damage to property, infrastructure, and flooding throughout this region [56,57]. Understanding spatial trends in TC intensity and landfall frequency and how these modulate the associated physical hazards in the NIO is very important for mitigating risk in these densely populated areas. While past studies have been somewhat hampered by the lack of observational data in the NIO, new satellite-based datasets now allow a more comprehensive investigation of TC behaviour without reliance on imperfect models, including a more detailed understanding of their wind and rainfall impacts after landfall. Furthermore, understanding historical patterns of TC behaviour in the region sets the groundwork for understanding how TC behaviour will change in the future.

The main aim of this study is to understand the detailed TC exposure in the NIO using geographical and environmental information. TC landfall variability over the NIO based on TC best track historical data during 1989–2018, and TC exposure, including wind and rainfall, over the NIO based on satellite-based observational datasets from 1998–2018 are investigated. The structure of the paper is as follows. Section 2 outlines the data and methodology used for this study. Results are presented in Section 3, and a summary and conclusions are provided in Section 4.

## 2. Data and Methodology

### 2.1. TC Data Information and Landfall Locations

Historical TC data in the NIO are sourced from version 4 of the International Best Track Archive for Climate Stewardship (IBTrACS). IBTrACS is maintained by the National Oceanic and Atmospheric Administration (NOAA) National Centres for Environmental Information [58]. It is a centralized repository for TC best track data from all TC warning centres globally and can be used to investigate the distribution, frequency, duration, size, and intensity of TCs throughout the world. For this analysis, the 3 h tropical cyclone location and maximum sustained surface wind (MSW) data from the New Delhi Regional Specialised Meteorological Centre (IMD) and the US Joint Typhoon Warning Center (JTWC) for the 30-year period from 1989–2018 in the NIO are used.

To obtain accurate information on TC landfall locations, the IBTrACS data are interpolated to 30 min temporal resolution. The TC centre positions are smoothly interpolated using cubic spline interpolation, and the maximum wind is linearly interpolated to conserve the maxima using the same methodology as that for the original IBTrACS data [59]. TC landfalls are then determined using the MATLAB landmask function.

ArcMap 10.7.1 was used to perform a spatial analysis of landfall locations and seasonal variations in TC landfalls. When a cyclone moves from ocean to land generally, that first inland TC placement is considered as the landfall point. The landfall points are then accumulated on a 25 km $\times$ 25 km resolution grid covering the NIO using the geoprocessing Fishnet and clip tools (ESRI 2011, ArcGIS Desktop). Then, using the clipping tool, combined TC landfalls and frequency-related TC information are retrieved in the form of a spatial geographical dataset within the NIO boundaries.

### 2.2. TC Data Record and Missing Intensity Values

The TC records in the IBTrACS database are not fully complete primarily because of missing intensity records, particularly prior to 1990. This is most likely because of two reasons: (1) many TCs were unobserved prior to the advent of satellites, and (2) IMD did not record intensity information in the best track records prior to 1990. In contrast, JTWC began recording intensity for some cyclones in the NIO in the early 1970s. Furthermore, in the IBTrACS database, IMD and JTWC have intensity estimates against different sections of

the total record. Thus, a simple methodology to develop a complete intensity record for TCs in the NIO has been developed for this study. First, the intensity record is examined for the IMD and JTWC databases separately and intercompared. The IMD database includes intensity information for 133 of the total 207 landfall events from 121 TCs out of 173 TCs, whereas the JTWC database includes intensity information for 149 of 207 landfall events from 119 TCs out of 173 TCs. A total of 75 TC intensity records from each of the two datasets overlap.

A Student's *t*-test is performed on the common records from the JTWC and IMD data to test whether the datasets for the Bay of Bengal and Arabian Sea region are significantly different. The test compares the average values of the common records to both datasets (Table S1) [60,61]. In all cases, the datasets are different at a 99% significance level ($p < 0.01$).

These differences are highlighted in Figure 2a, which shows box and whisker plots for each of the IMD and JTWC intensity data separated into the Bay of Bengal and the Arabian Sea. The six number labels that are shown in Figure 2a are the minimum, first quartile, median, third quartile, maximum and the out of the range maximum number. The two distributions clearly differ depending on the sub-basin region. In the Bay of Bengal, JTWC intensity data (grey) are higher than IMD data (orange) and have a higher spread, whereas in the Arabian Sea, although the IMD (green) mean is higher than JTWC (purple), the two distributions exhibit similar variance. To maximise the amount of intensity data available to the study, a simple methodology to re-scale the JTWC data to IMD data is developed. Using the records that are common to both datasets for the Bay of Bengal and the Arabian Sea, a linear regression equation is calculated that relates the two datasets. This is then applied to the sections of the JTWC data that are missing in the IMD data to obtain the Combined Intensity Dataset (Figure 2b). After applying the linear regression equation (Table S1), the distribution of the combined intensity dataset for the Bay of Bengal (blue) and the Arabian Sea (gold) is shown in Figure 2a.

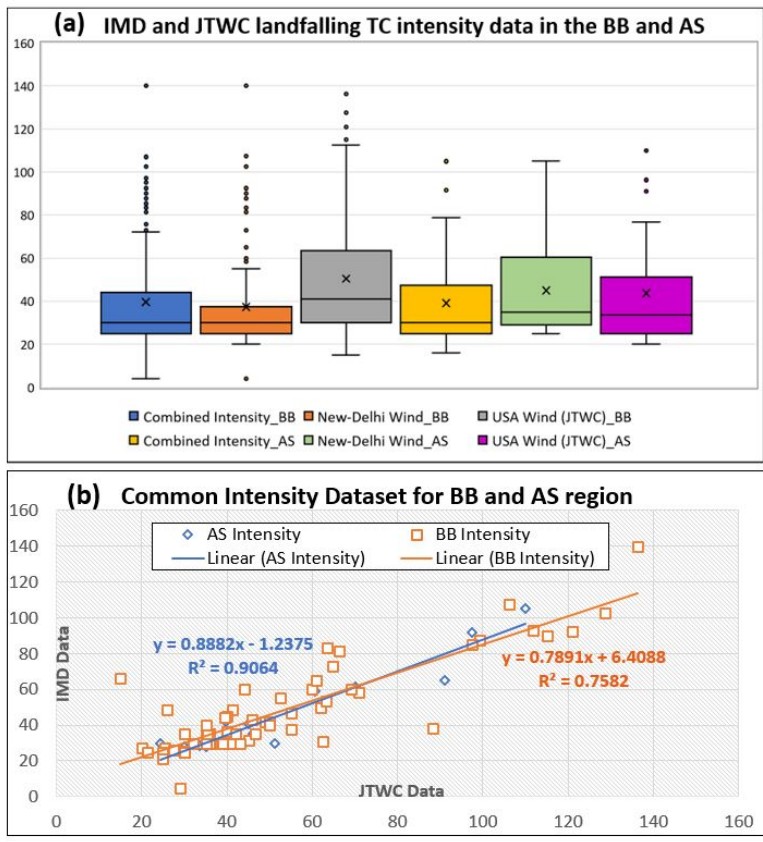

**Figure 2.** (**a**) Intensity data for the Bay of Bengal (BB) and the Arabian Sea (AS) region using the Indian Meteorological Department (BB-orange, AS-green), Joint Typhoon Warning Center (BB-grey,

AS-magenta), and the Common dataset (BB-blue, AS-gold); and (**b**) linear regression and r-squared values for the Common Intensity Dataset for the BB (blue) and AS (orange) regions. In panel (**a**) the mean is indicated by "x" and outlier points are indicated by filled circles.

### 2.3. The Indian Ocean Dipole

The Dipole Mode Index (DMI), which is the difference in average SST anomaly between the western ($0°$ E$-70°$ E and $10°$ S$-10°$ N) and eastern ($90°$ E$-110$ $°$E and $10°$ S$-0°$ N) tropical the Indian Ocean is used to determine the IOD phase and is available from the Bureau of Meteorology (http://www.bom.gov.au/climate/iod/, accessed on 1 August 2022). Generally, the positive IOD phase begins developing during the boreal summer and peaks in boreal fall [31]. During the period of study, there were 18 neutral, 7 negative and 5 positive IOD years.

### 2.4. Atmospheric Data

Atmospheric data from the European Centre for Medium Range-Weather Forecast (ECMWF) fifth-generation reanalysis (ERA5) [62] are used to represent the environmental conditions associated with the pre-, post-, and monsoon seasons and also by the IOD phase [63]. The temporal resolution of the dataset is three-hourly, although the monthly average data was downloaded for this study and the variables that are analyzed are geopotential height and wind fields for 1989 to 2018 over the region of interest.

### 2.5. TC Induced Rainfall Distribution

Tropical Rainfall Measuring Mission (TRMM) 3B42 V7 multi-satellite satellite data [64,65] in the NIO region during the 21-year period of 1998–2018 are used to analyze the TC rainfall contribution for the study period. Note that the period for the rainfall analysis is shorter than for landfalling behaviour. The TRMM 3B42 provides three-hourly data with a resolution of $0.25° \times 0.25°$ latitude/longitude. In agreement with previous studies [66–68], rainfall events that occurred within 500 km radius of the TC centre are categorized as TC-induced rainfall because this area falls within the maximum extent of the wind circulation (80–400 km) and the range of the curved TC cloud shield (a 550–600 km radius) [69,70]. Because of the lack of a consistent rainfall dataset prior to 1998, the TC-induced rainfall is calculated from 1998 to 2018.

### 2.6. TC Wind Fields—The DAV Wind Parameter Dataset

The spatial wind field parameters started being recorded by JTWC in IBTrACS beginning in 2002, although many records are missing after that time. Thus, the DAV technique, which can be used for the entire infra-red (IR) satellite record, is used to develop a TC wind radii dataset for the NIO [71,72] that extends back to 1998 to match the TRMM rain set. The DAV technique in [71,73] is based on the concept that TC clouds become more axisymmetric and organized as a TC intensifies, and therefore cloud structure can be used to estimate TC intensity. Briefly, cloud brightness temperature gradients are calculated for each image pixel, and the deviation between the gradient and a perfectly axisymmetric vortex is calculated (the deviation angle). The variance of all deviation angles then provides a quantitative measure of cloud axisymmetry relative to a single reference pixel. DAV maps are created by treating each pixel, in turn, as a reference pixel. Pixels with low DAV values indicate high TC intensity, while high DAV values indicate low intensity.

For this study, DAV maps were created for the period 1998 to 2018 using images from the NOAA/CPC half-hourly global merged IR satellite product [74]. The base satellite imagery in the Indian Ocean for this product are longwave (10.8 μm) IR images from the European Agency (Meteosat-8) satellite positioned at 41.5° E. The DAV maps were azimuthally averaged every 10 km from the TC centre radially out to 600 km and smoothed using a 24 h backward running mean to match the smoothness of the best track wind radii [71]. Next, the distance from the TC centre to the DAV values that best represent each of the R34, R50, and R64 best track wind radii [71] are extracted from each azimuthally-

averaged DAV map and used as predictors in a wind radii regression model, which also uses SST, TC age (the time since the TC first reached 34 kts intensity), and the TC maximum wind. Both the DAV "values" for each wind radii and the coefficients for the regression model were derived using the North Atlantic dataset [71]. The regression model was run to produce estimates of R34, R50, and R64 for every record in IBTrACS that has a maximum wind of at least 34 kt, 50 kt, and 64 kt, respectively. The DAV wind radii were then used to create maps of TC wind exposure in the NIO for winds of at least 34 kts, at least 50 kts, and at least 64 kts from 1998–2018.

## 3. Results

### 3.1. TC Activity and Landfalling Trends in the North Indian Ocean

During the study period, a total of 282 TCs, including tropical depressions, formed in the NIO, with an average of 9.1 TCs per year. Of these, 219 TCs formed in the Bay of Bengal and 63 formed in the Arabian Sea. Tropical depressions are included in the statistics in this study, which explains the higher annual count compared with [18,19], because of their known impacts upon landfall, especially due to heavy rainfall [75,76]; Figure 3 shows the number of TCs that form in the NIO along with the number of TCs that make landfall at least once in the NIO over the 30-year period annually and by month. There is considerable inter-annual variability during the period of study, with peaks in activity approximately every 12 years, possibly corresponding to an unknown climatic signal. While there is an overall decreasing tendency in the number of TCs that form in the NIO over the study period, it is not significant.

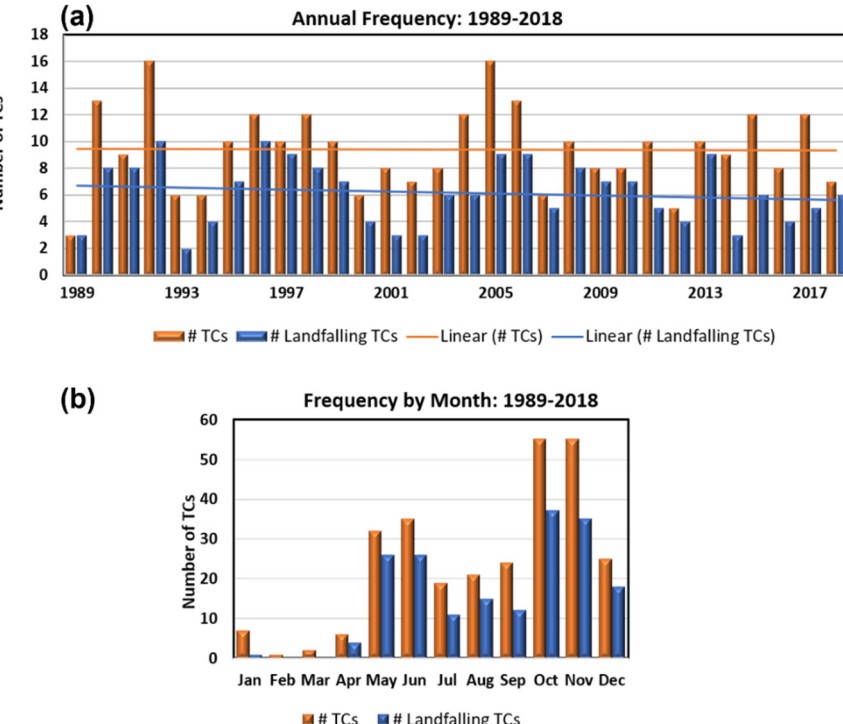

**Figure 3.** Histogram of the number of TCs that form (orange) and make landfall (blue) in the NIO from 1989–2018: (**a**) annual frequency and (**b**) by month. "#" refers to "number" in both panels.

Approximately 66% (185) of the TCs that formed in the NIO made landfall during the period, with 155 (84%) TCs making landfalling in the Bay of Bengal and 30 (16%) in the Arabian Sea. There are many more landfalls in the Bay of Bengal than in the Arabian Sea because of the larger number of TCs that form in the Bay of Bengal and the close proximity of land in that sub-basin. In addition, there is a higher density of TC tracks that made landfall in the upper northwest of the Bay of Bengal, particularly along the Kolkata, Hyderabad, Madras and Bangalore coastlines, which are highlighted in the higher-

frequency red and pink pixels in Figure 1b and also by the higher track density region in red in Figure 1c. Landfall frequency in the northeast Bay of Bengal region is relatively less dense. Several TCs in the NIO make landfall multiple times. For example, a TC can make landfall in Sri Lanka, and India from the Bay of Bengal, and then move offshore over the Arabian Sea, recurve, and make landfall again on the west coast of India (Figure 1a). As a result of these multiple landfalling TCs, a total of 222 actual landfalls are recorded during the period of study. Furthermore, because of the close proximity of the NIO to the western North Pacific, TCs that make landfall in the NIO can form in the Western North Pacific Ocean and then track westward across Southeast Asia and into the NIO. Figure 1a illustrates this with 10 TCs entering the region from the eastern boundary of the figure during the period of study. The predominant tracks are westward or recurving north to northeast. Sixty-three percent of the TCs that either form in the NIO or track into the NIO from the east make landfall in this region (Figure 1a).

### 3.2. Seasonal Distribution of TC Landfall

Figure 3b shows the distribution of TCs and TC landfalls by month during the 30-year study period. In general, the pattern of landfalling TC matches the overall activity. During this period, no TC made landfall during February and March in the NIO. Landfall frequencies tend to peak during the transition between the late pre-monsoon (May) and monsoon periods (June–September), with high landfalls in May and June. Landfall frequencies also peak during the post-monsoon periods (October–January) (Figure 3b). While landfalls do occur during the peak monsoon period, they are considerably lower than during the pre and post periods.

The spatial variation of landfalling TCs in the NIO region separated by Monsoon season and the 30-year mean deep layer (200–850 hPa) steering flow associated with the pre-monsoon, main monsoon, and post-monsoon seasons are depicted in Figure 4. The individual flow patterns at 200, 500, and 1000 hPa are provided in Figure S1.

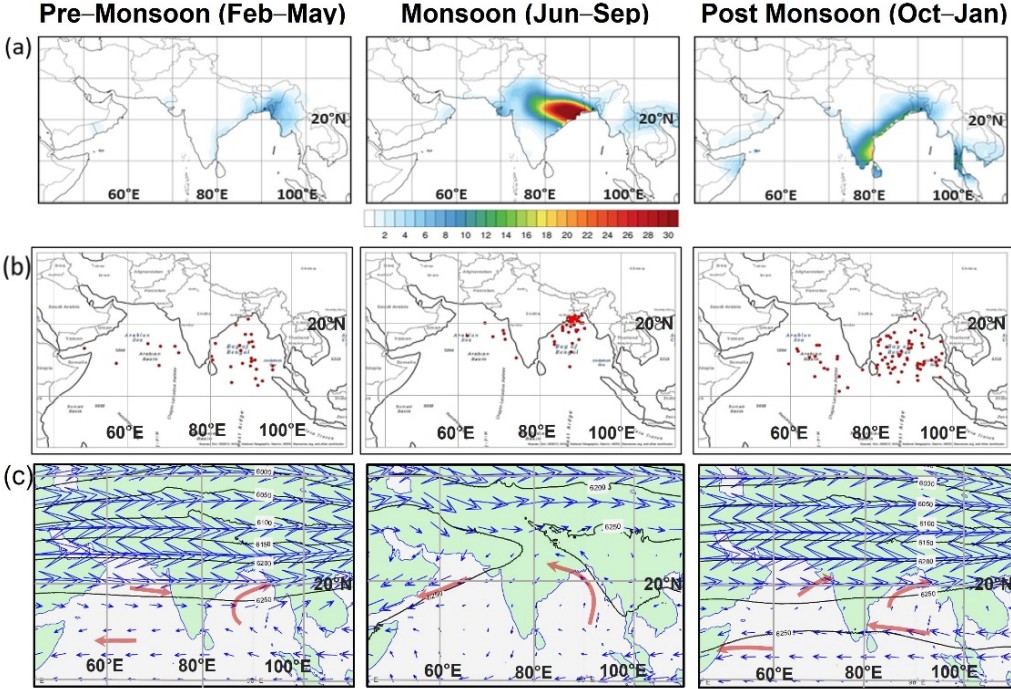

**Figure 4.** Patterns separated by Monsoon phase: (**a**) landfalling TC track density; (**b**) TC genesis locations indicated by red dots; and (**c**) 30-year average 200–850 hPa deep layer steering flow (blue vectors) and 500-hPa geopotential heights (black contours) for the pre-monsoon (left column), monsoon (middle column), and post-monsoon (right column). Red arrows in panel (**c**) indicate the prevailing steering flow.

During the pre-season (February–May), formation locations are spread out across the Arabian Sea and the Bay of Bengal. The deep layer environmental steering flow is generally strong westerly north of 15° N and will cause the few TCs that develop in the Arabian Sea to move to the east, making landfall on the east side of the basin. In the Bay of Bengal, the flow is weak easterly in the southern portions of the Bay, switching to southerly and more strongly westerly in the northern part of the Bay. This flow steers TCs to the north and east, causing TC landfall to occur preferentially along the northeast coastline of the Bay of Bengal (Figure 4a,b left column).

During the main monsoonal period from June to September (Figure 4 middle column), the deep layer mean steering flow switches to a north-easterly weak offshore flow dominated by the upper-level easterly jet on the south side of the Tibetan high-pressure system (Figure S1) in the northern part of the Arabian Sea, which increases in strength on the western side of the Arabian Sea. However, the surface (1000 hPa) flow is strongly onshore, as would be expected during the peak monsoon, and genesis locations are clustered on the eastern side of the Sea. Thus, cyclones are generally directed onshore toward the coastal regions of Gujarat in northern India, with some that form in the western part of the Arabian Sea steered into Oman by the stronger deep layer mean flow just off Oman. In the Bay of Bengal, TCs cluster in the north of the Bay of Bengal and are steered northward and westward by the deep layer steering flow leading to a preferential landfalling area on the northwest side of the basin into north-eastern India and southwestern Bangladesh (Figure 4). These landfalling TCs also typically have much longer inland tracks to the west after landfall compared with during the pre- and post-monsoon periods, likely because of the much weaker westerly deep-layer steering flow over land during the monsoon period compared to the pre- and post-monsoon periods (Figure 4c).

During the post-monsoon period from October to January (Figure 4), the mean deep layer steering flow in the southern part of the North Indian Ocean switches to easterly while the flow in the northern part of both the Arabian Sea and the Bay of Bengal is westerly. In the Arabian Sea, TCs are steered into the Horn of Africa, while TCs are generally steered along the entire western to the northeastern coastline of the Bay of Bengal. TCs moving west from the South China Sea are also steered across the Thailand peninsula during this season.

*3.3. Spatial Shift of TC Landfalling Tracks*

Although no statistically significant trend in the number of landfalling TCs over the 30-year period in the NIO was found in Section 3.1, there is a spatial shift in the locations of landfalling TCs with a decrease in recent years of landfalls on the west coast of India, Myanmar, and Thailand, and an increase in landfalls along the southeast coast of India, Sri Lanka, the northeast coast of India, and into Bangladesh (Figure 5). The results are relatively insensitive to the period of time the change is calculated over (e.g., Figure 5c,d). This suggests that there are regions in the Bay of Bengal, including the east coast of India and Bangladesh, that are experiencing higher TC exposure because of a shift in tracks rather than because of more frequent TC formation.

Figure S2 shows that the majority of the spatial shift occurs during the peak monsoon season, with an increased frequency of landfalls occurring in northeastern India and Bangladesh and then spreading inland across northern India (Figure S2c). There are also some shifts during the post-monsoon with a decrease in landfalling TCs on the southwest coast of India and an increase on the southeast coast of India and Sri Lanka in the latter 15-year period. A decrease in landfalls in Myanmar and across the Thailand peninsula is also attributed to a change in post-monsoon landfalling tropical cyclone behavior (Figure S2d). While these changes are likely due to shifts in the overall monsoon circulation over the past 30 years, the specific mechanisms are beyond the scope of this study.

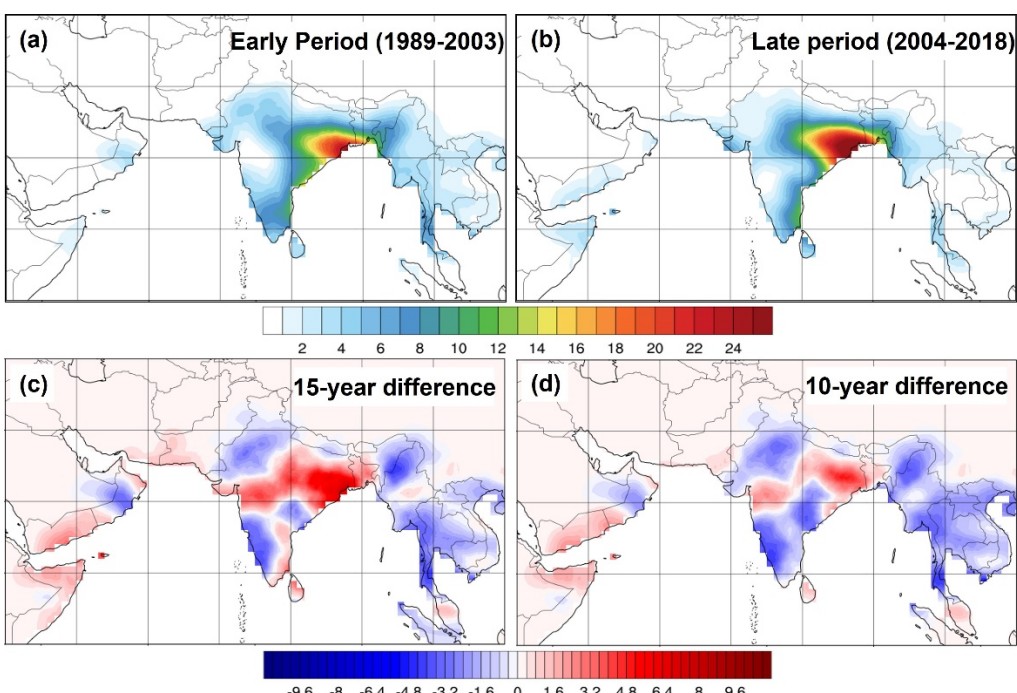

**Figure 5.** Landfalling TC Track Density for (**a**) the Early Period (1989–2003); (**b**) the Late period (2004–2018); (**c**) the difference between the Early and Late periods; and (**d**) the difference calculated using 10-year periods 2009–2018 minus 1989–1998 to demonstrate the sensitivity to periods chosen. Density Maps are calculated at 0.1° resolution.

### 3.4. Influence of the Indian Ocean Dipole (IOD) on Occurrence and Landfalls of TCs

Following the work by Yuan and Cao [30], the number of landfalling TCs from 1989 to 2018 is separated into the three IOD phases (Figures 6 and 7). There are relatively few positive (5) and negative IOD (7) years compared to neutral IOD (18) years during the period of study. Average annual numbers of TCs that form in the NIO are similar during positive, negative, and neutral phases (9.1–9.3). However, on average, slightly fewer TCs make landfall during neutral IOD phase years (6.0) compared to positive and negative phase years (6.4) (Figure 6). There is high variability in the annual number of TCs that make landfall during the negative IOD phase, with as many as 10 and as few as 3 making landfall (Figure 6).

In order to intercompare landfalling track patterns, the track densities are normalised by the number of years in each IOD phase in Figure 7. The spatial pattern of landfalls during the negative IOD phase (Figure 7a) is quite evenly dispersed, especially in the Bay of Bengal, compared with the neutral and positive phases. Similarly, genesis locations are more dispersed in negative phase years, whereas genesis locations in positive years tend to cluster in the northern part of the Bay of Bengal (Figure 7b). The neutral phase exhibits both some clustering of genesis locations in the north of the Bay of Bengal and some more evenly dispersed genesis locations. The deep layer mean (DLM) steering flow looks very similar in all phases (Figure 7c), with the weak easterly flow in the southern part of the Bay of Bengal and weak southerly steering flow in the northern part of the Bay. The preferred landfalling locations in each of the three phases are more strongly modulated by the differences in genesis locations of the TCs than in the small differences in steering flow with TCs in neutral and positive IOD years steered directly into northern India and Bangladesh and those in the negative phase more evenly steered onto the Bay of Bengal coastline (Figure 7a).

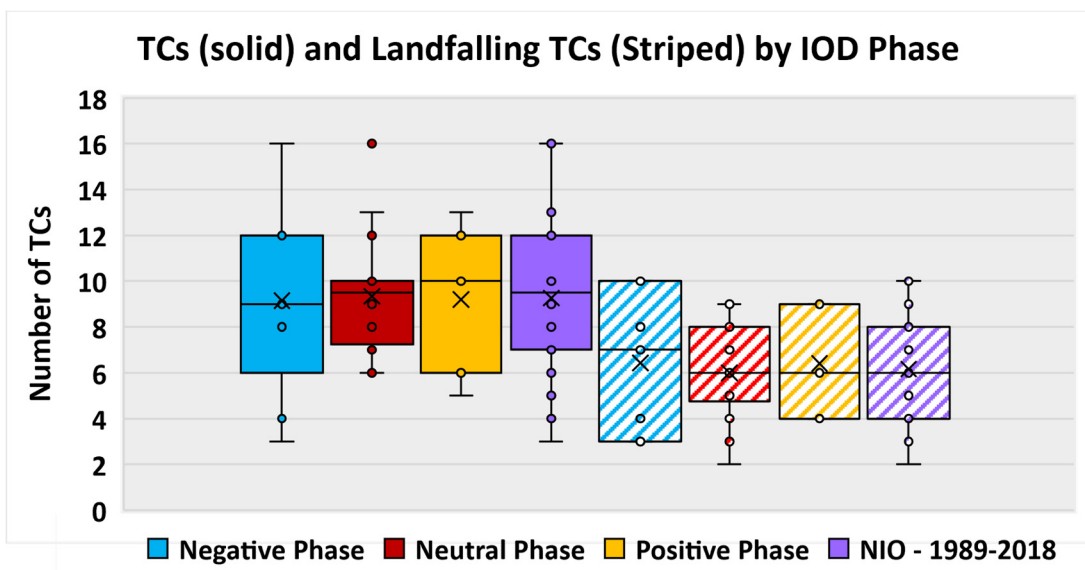

**Figure 6.** Box-and-whiskers plots of annual TC activity in the NIO region separated by the Indian Ocean Dipole (IOD) phase for all activity (solid) and landfalling activity (striped) for the negative phase (blue), neutral phase (red), positive phase (gold), and all TCs (purple). The mean is indicated by "x" and all data points are indicated by open circles.

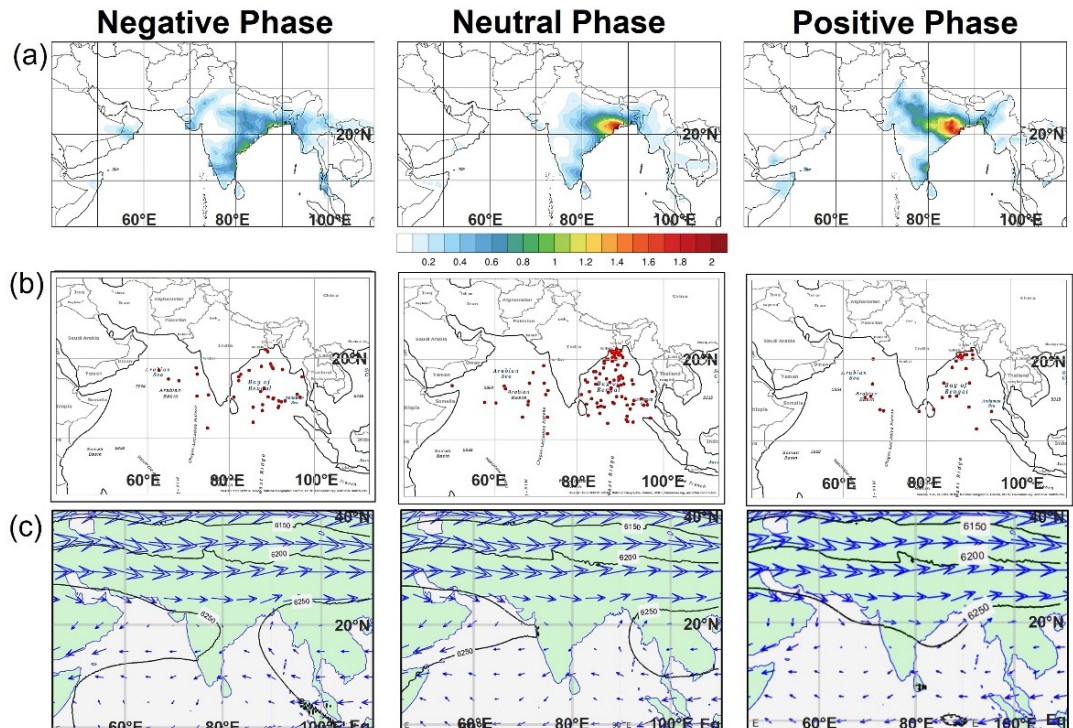

**Figure 7.** Patterns separated by IOD phase: (**a**) average annual landfalling TC frequency; (**b**) TC genesis locations indicated by red dots; and (**c**) 200–850 hPa deep layer steering flow (blue vectors) and 500-hPa geopotential heights (black contours) averaged from May to November (1989–2018).

*3.5. TC Intensity at Landfall*

The intensity of a TC is generally measured by the maximum sustained 10 m wind speed (Vmax). High winds are a major factor in producing damage after a TC makes landfall. These damages are highly correlated with both the intensity and the wind field size of a tropical cyclone [77].

TC landfalling frequency by the intensity in the Bay of Bengal and the Arabian Sea using the combined intensity dataset is shown in Figure 8a. Because of the simplicity of the regression methodology to combine the IMD and JTWC datasets, TC intensity is categorized using the JTWC categories: tropical depression (TD) (Vmax < 33 knots), tropical storm (TS) (34 knots ≤ Vmax < 63 knots), typhoon (TY) (64 knots ≤ Vmax < 130 knots), and super typhoon (STY) (Vmax ≥ 130 knots). The combined intensity dataset includes 97% of the records, with only 3% (shown in the orange bar in Figure 8a) unable to be recovered through the regression technique. In the Bay of Bengal, the intensity of TCs at landfall is heavily skewed to the weakest (TD) intensities (57%) (Figure 8). However, at least 13% made landfall as a typhoon or super typhoon, and 25% made landfall as a tropical storm. In the Arabian Sea, the intensity of TCs at landfall is also heavily skewed to tropical depressions (62%), with 21% making landfall as a tropical storm and 14% making landfall as a typhoon. The number of landfalls by intensity category each decade suggests that all TC intensity categories except for tropical depressions have decreased over the 30-year period (Figure 8b). Only one TC made landfall in the NIO as a super typhoon in the 30-year period, and this occurred in 1999.

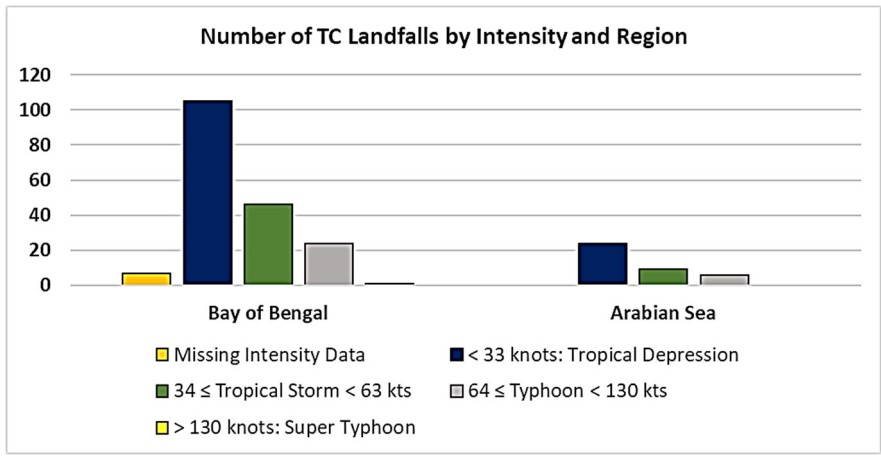

(**a**)

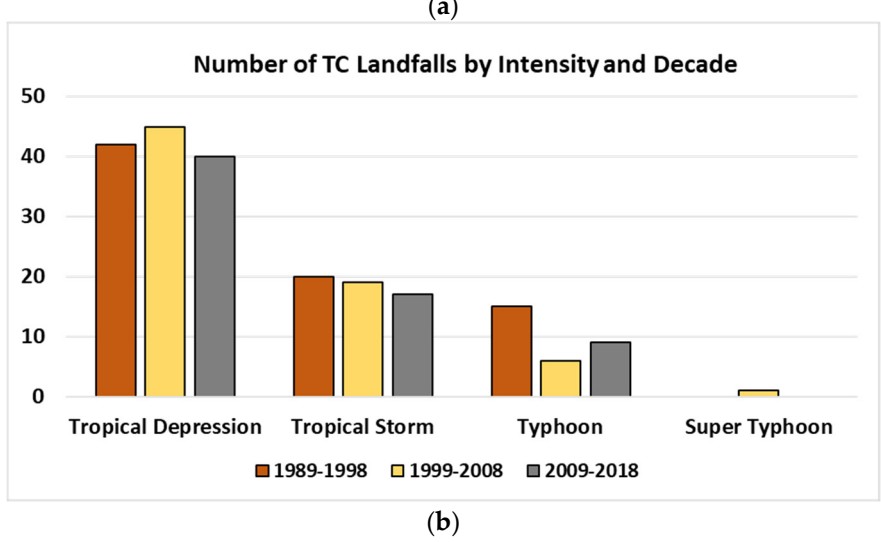

(**b**)

**Figure 8.** Histogram of: (**a**) the number of TC landfalls in the Bay of Bengal and Arabian Sea by JTWC intensity scale; and (**b**) the number of TC landfalls in each 10-year period separated by JTWC intensity category.

The spatial distribution of TCs by the intensity at landfall is shown in Figure 9. Figure 9 suggests that TCs preferentially make landfall at TD intensity in the west region of the Bay of Bengal (Kolkata) whereas TCs that make landfall at TS and TY intensity, although

fewer, tend to be more evenly distributed around the coastal regions of the Bay of Bengal. The only TC to make landfall as a STY made landfall in the exposed southeast coastline of Kolkata during the 30-year period.

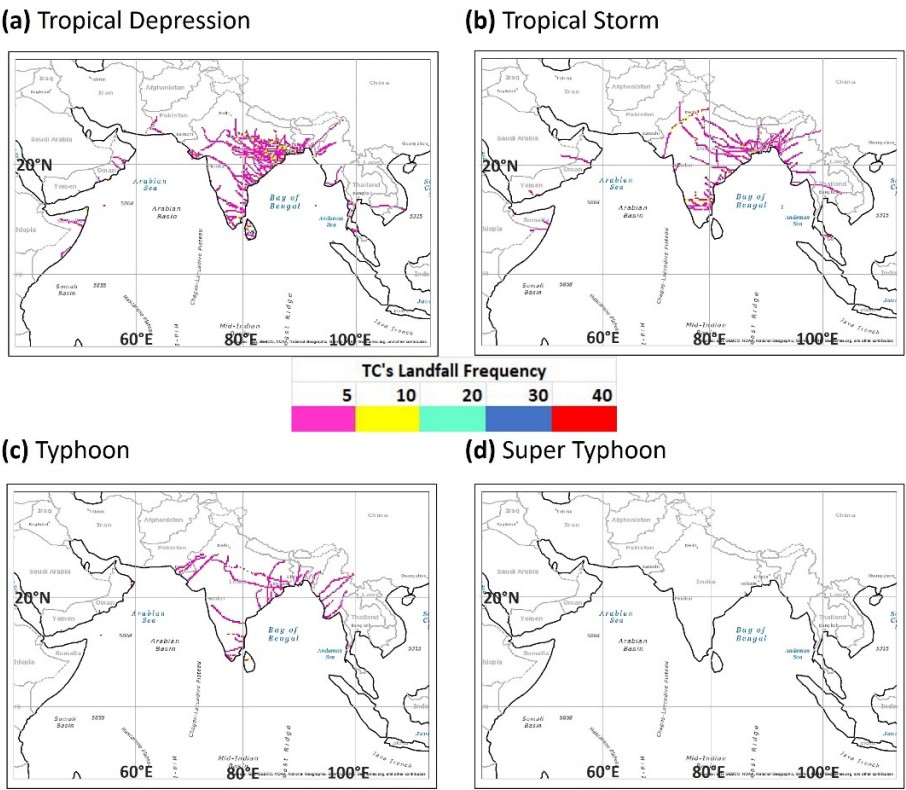

**Figure 9.** TC landfall frequency separated into intensity categories using the combined wind dataset: (**a**) tropical depression; (**b**) tropical storm; (**c**) typhoon; and (**d**) supertyphoon.

### 3.6. Distribution of the TC Surface Winds and Rainfall after Landfall

The spatial distribution of the TC-related wind field and accumulated rainfall upon landfall is shown in Figure 10. Most of the coastal regions in both the Arabian Sea and the Bay of Bengal are impacted by at least 34 kt winds due to TCs during the 20-year period and, in some regions, at least once a year (Figure 10a). Along the coastline of the Bay of Bengal, the frequency is much higher, and much of the coastline is impacted 2–3 times per year by TC-related winds of at least 34 kts. In addition, the southwest coastline of the Bay of Bengal, including Sri Lanka, is frequently impacted by winds of at least 50 kts, which are considered a threshold for damage (not shown).

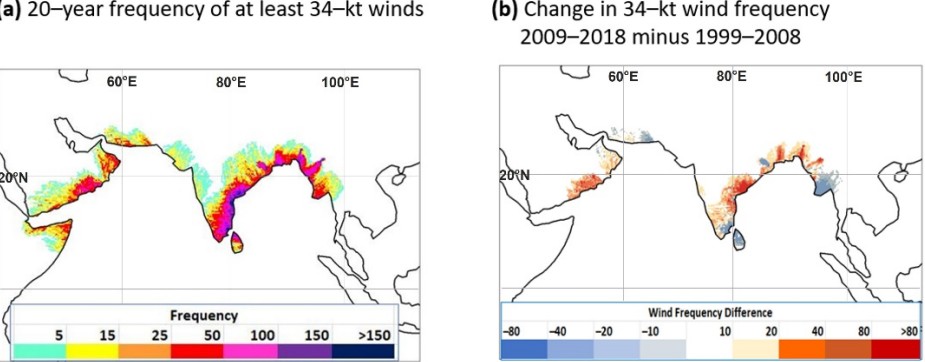

**Figure 10.** Frequency of at least 34 kt winds after landfall based on the DAV wind radii for (**a**) 1999–2018; and (**b**) change in frequency from the first to the second decade.

Similar to the shift in TC track frequency (Figure 5), the frequency of TC winds of at least 34 kts affecting coastal and inland regions has shifted over the past 20 years such that the frequency has increased along much of the coastal regions of both the Bay of Bengal and the Arabian Sea. In particular, the east coast of India, Bangladesh and northern Myanmar, Oman and the Horn of Africa have increased frequency of 34 kt winds, while Sri Lanka and small regions of India and Myanmar have seen a decrease in the frequency of at least 34 kt winds associated with TC landfall (Figures 10b and S3).

The highest TC rainfall amounts tend to be slightly skewed to the eastern side of the Bay of Bengal such that the coastal regions of Bangladesh, Myanmar and Cambodia receive the highest total rainfall accumulations in the NIO (Figure 11a). In the Arabian Sea, there is a similar pattern, with the east side of the basin, including the west coast of India, receiving the highest rainfall. The TC rainfall comprises between 5 and 25% of the total annual rainfall in most regions except for the coastal regions of Oman, where TC rainfall can reach up to 50% of the total annual rainfall (Figure 11c). The difference between the two 10-year periods, 1999 to 2008 and 2009 to 2018, are shown in Figures 11d and S4, highlighting the decrease in rainfall due to TCs that has occurred over the 20-year period across most of the subcontinent except Sri Lanka, the Thailand peninsula, Malaysia, and Indonesia. This is in spite of the shift of TC tracks further north in the Bay of Bengal (Figure 5). TC-related rainfall has also increased in the latest 10 years over the countries bordering the western half of the Arabian Sea (Figure 11d). Finally, during the pre-monsoon the rainfall amount tends to concentrate over the ocean region, with lower amounts of rainfall evenly distributed along the coastal areas (Figure S5). During the monsoon season, the rainfall is highly concentrated on the eastern side of both the Arabian Sea and the Bay of Bengal regions. In the Bay of Bengal, the rainfall is concentrated along the west coast of Myanmar from Bangladesh to Thailand and during the post-monsoon period, the rainfall shifts further south over Indonesia and the Malaysia peninsula (Figure S5).

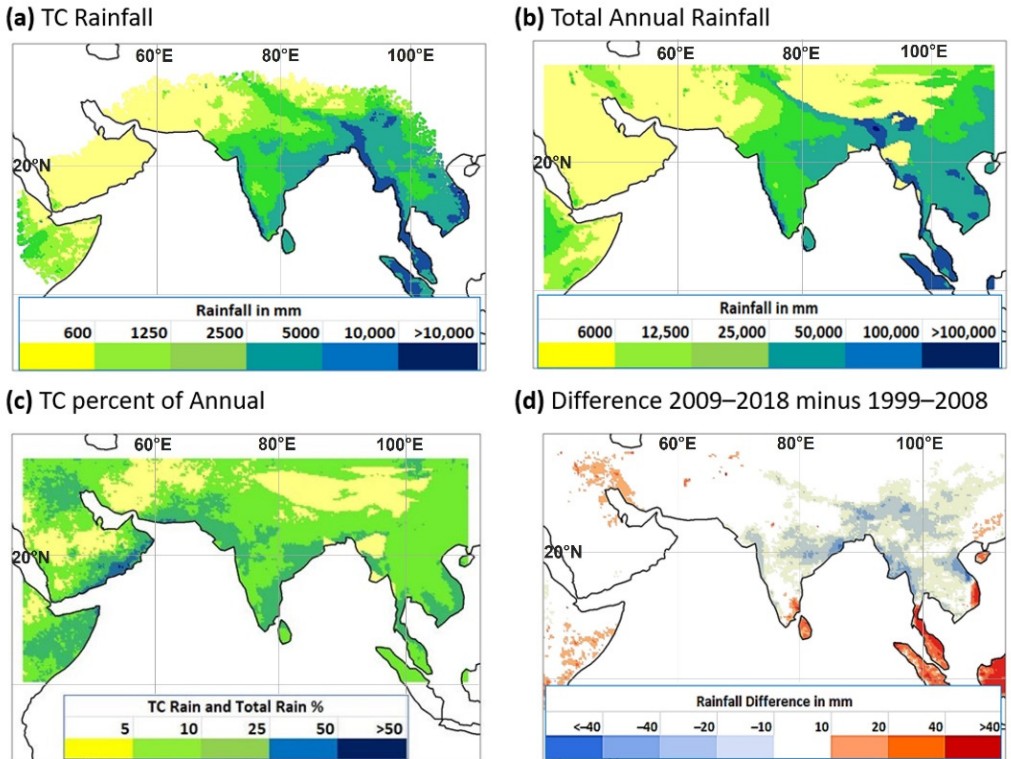

**Figure 11.** Extent of over land rainfall (mm) for the 20 years from 1999 to 2018 for (**a**) TC rainfall; (**b**) Total annual rainfall; (**c**) percent TC annual rainfall; and (**d**) the difference between the two 10-year periods 1999–2008 and 2009–2018.

Compared with the winds, the rainfall extends quite far inland (Figures 10a and 11a). This is not surprising since the surface winds associated with landfalling TCs erode quite rapidly due to the increased friction of the land surface compared with the oceans. However, the TCs continue to track inland and winds blow; the TC threshold and associated rain can, therefore, also extend quite far. The TC track length, duration, and speed inland after making landfall as recorded in the combined best track dataset is calculated from the landfall location along the coast of the Arabian Sea and Bay of Bengal to the end of the recorded track in the combined dataset. Over the 30-year period, the mean TC inland track length, duration, and speed in the Bay of Bengal have decreased (Table 1). A similar trend is observed in the Arabian Sea for length and duration, although because of the much smaller number of TCs there, the result is not as robust. In general, the inland track length and duration are considerably shorter in the Arabian Sea compared with the Bay of Bengal (Tables 1 and 2). During the peak monsoon period (June–September), TC inland tracks in the Bay of Bengal are longer, and the average duration is higher (Figure 12, Table 2) compared with the pre- and post-monsoon seasons. Interestingly, although the average speed of motion is considerably higher in the Bay of Bengal than in the Arabian Sea during the pre-monsoon period, they are comparable during the monsoon and post-monsoon periods.

**Table 1.** Mean inland track length (km), duration (days), and speed (km/hr) by decade for the Bay of Bengal and the Arabian Sea.

| | Bay of Bengal | | | Arabian Sea | | |
|---|---|---|---|---|---|---|
| | 1989–1998 | 1999–2008 | 2009–2018 | 1989–1998 | 1999–2008 | 2009–2018 |
| Maximum Track Length (km) | 642.0 | 525.2 | 392.4 | 346.0 | 179.1 | 266.9 |
| Mean Duration (days) | 1.87 | 1.53 | 1.23 | 1.07 | 0.47 | 0.82 |
| Speed (km/hr) | 17.8 | 18.6 | 15.3 | 14.5 | 14.2 | 14.3 |

**Table 2.** Inland TC Length Duration and Speed after Landfall separated by monsoon period.

| | Bay of Bengal | | | Arabian Sea | | |
|---|---|---|---|---|---|---|
| | Pre-Monsoon | Monsoon | Post-Monsoon | Pre-Monsoon | Monsoon | Post-Monsoon |
| Maximum Track Length (km) | 909 | 2141 | 1061 | 703 | 829 | 812 |
| 30-year mean track length | 413 | 815 | 336 | 181 | 392 | 250 |
| Maximum Duration (days) | 2.1 | 8.0 | 6.3 | 2.4 | 1.8 | 3.3 |
| 30-year mean Duration (days) | 0.8 | 2.4 | 1.2 | 0.6 | 1.0 | 0.9 |
| Maximum Speed (km/hr) | 46.5 | 41.9 | 56.5 | 14.5 | 25.7 | 37.5 |
| 30-year mean Speed (km/hr) | 23.0 | 16.5 | 16.1 | 13.0 | 15.2 | 14.3 |

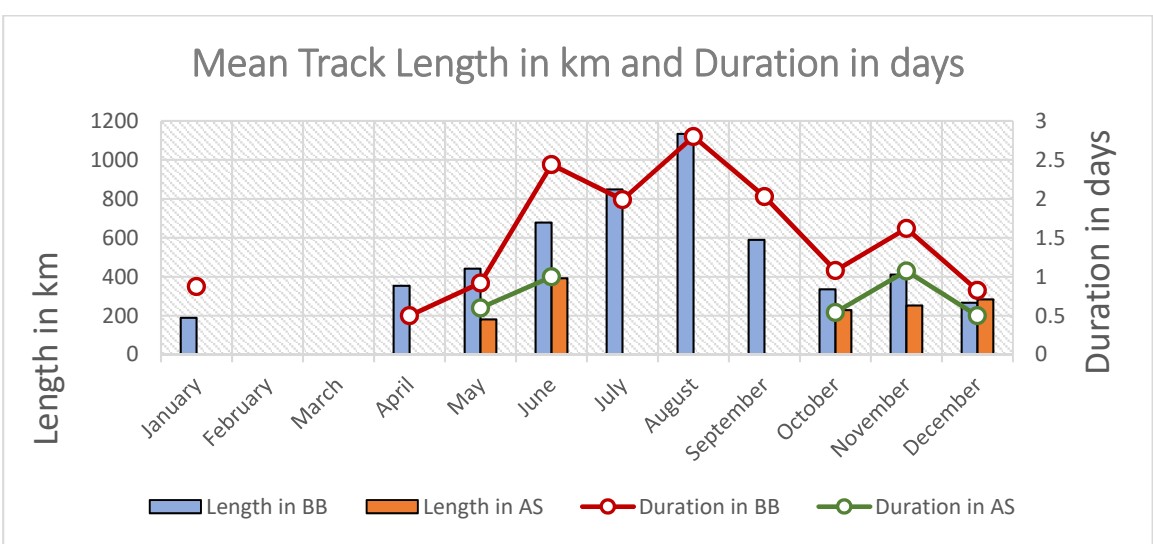

**Figure 12.** Monthly variation of TC over land mean track length and duration for the Bay of Bengal (BB) and the Arabian Sea (AS) region.

## 4. Summary and Conclusions

This study presents a TC landfall climatology over the NIO region during the period 1989–2018 and associated wind field and rainfall climatology during 1998–2018. Landfalling TCs and their intensity characteristics are determined using the IBTrACs version 4 dataset, which is a compilation of global tropical cyclone Operational Center Best Track archives. The ERA5 atmospheric reanalysis data, TRMM satellite data, and the satellite-derived DAV wind data are also used to investigate the physical characteristics of TCs in the NIO, especially after landfall. This study analyzed the landfalling TC trends over a 30-year period, the impact of the IOD phase on landfalling TCs, TC intensity variation, landfalling TCs, and their inland track patterns, TC induced rainfall amount, monsoon seasonal variations in TC landfalls, including monsoon-scale drivers and steering flow analysis for the NIO region. In addition, the landfalling TCs' inland track length, duration, and speed after final landfall have also been studied.

A total of 282 TCs formed in the NIO during 1989–2018, and of these, approximately 66% made landfall, with 155 TCs making landfall in the Bay of Bengal and 30 TCs making landfall in the Arabian Sea over the 30-year period. The number of landfalling TCs in the NIO has decreased over the 30-year period, although the trend is weak. There is also an interannual variability with peaks in activity approximately every 12 years over the 30-year period, which do not appear to correspond to known climatic signals.

The majority (~130) of all TCs make landfall as tropical depressions. However, approximately 59 made landfall as tropical storms, and approximately 10 made landfall at typhoon intensity.

The seasonal variability of TC activity in the NIO has been well documented [13,14,39,42], with low activity in the pre-monsoon, high activity in the early monsoon (June) but lower activity during the main monsoon (July–September), and much higher activity during the post-monsoon period. The landfall activity mimics this generally NIO activity. Moreover, the spatial variation of landfalling TCs in the NIO region changes with the season, with the majority of landfalling activity in the pre-monsoon occurring in northern Myanmar and eastern Bangladesh, increased landfalling activity in northeast India and Bangladesh during the main monsoon period with long westward tracks inland, and activity spread evenly around the Bay of Bengal during the post-monsoon with higher landfalling activity on the southeast coast of India and Sri Lanka and also in Thailand. Despite relatively lower TC activity during the monsoon period relative to the pre- and post-monsoon periods, higher accumulated TC-related rainfall tends to fall during the monsoon period, particularly along

the eastern coastlines of the Arabian Sea and the Bay of Bengal. Higher TC-related rainfall also extends further inland in India, Bangladesh and Myanmar during the monsoon season.

The landfalling activity has shifted over the 30-year period with decreased landfalls on the southwest coast of India, increased landfalls on the northwest and southeast coasts of India and Sri Lanka, and the northeast coast of India into Bangladesh. Landfalls have decreased in Myanmar and Thailand. Furthermore, the frequency of TC-related winds of at least 34 kts has increased across much of the region, particularly in Bangladesh, Oman and the Horn of Africa. However, the TC-related rainfall has generally decreased over most of the region except for Sri Lanka, the Thailand peninsula, Malaysia, Indonesia, and the countries bordering the west side of the Arabian Sea, in contrast with studies that suggest that generally, TCs will become rainier as the climate warms [8].

Whereas TC genesis decreased (increased) during the positive (negative) phase of the IOD [30], we found that similar numbers of TCs (9.1–9.3 per year) formed during all phases of the IOD, perhaps because we included tropical depressions in the numbers. We felt it important to include tropical depressions since even these relatively weak systems can have large impacts through heavy rain upon landfall. We also found that landfall tended to occur on the northwest coastline of the Bay of Bengal during positive IOD years in contrast to [30], who hypothesized that landfalls would be more likely to occur on the eastern coastline of the Bay of Bengal due to increased upper-level westerlies during positive IOD phases. When normalized by the number of years, we found that positive and neutral IOD phases had very similar patterns of landfall with concentrated landfalls onto the northeast coast of India and into Bangladesh. During the negative phase of the IOD, landfall tended to occur more evenly spread around the entire Bay of Bengal coastline.

The NIO region is one of the most vulnerable regions to TC-induced risks worldwide. While this study suggests that there may be a slightly decreasing trend in the frequency of landfalling TCs in the region, there is a shift in the spatial pattern of landfalling TCs, with a higher number of landfalling TCs occurring across northern India and Bangladesh, creating additional exposure to TC-related hazards in those regions. This is similar to a shift in TC exposure documented for the western North Pacific [78] and Southeast Asia [79] regions and highlights the importance of understanding the fundamental changes in TC behavior that are already occurring regionally in TC-prone areas. Studies such as this add to the rapidly expanding body of knowledge on the physical impacts of TC, which impose long and short-term impacts on the livelihoods, economy, agricultural sector, fisheries, and marine life, and also create risk for these sectors. The research in this paper is based on new satellite-based records that have recently reached a 20-year length making them more suitable for long-term studies of TC exposure. Further increasing the length of these datasets will enable us to better understand the long-term changes in TC exposure in this unique region. Future work includes using numerical models to produce climate-scale simulations of TC behavior to better understand the past, present, and future behavior of TCs in this region.

**Supplementary Materials:** The following supporting information can be downloaded at: https://www.mdpi.com/article/10.3390/atmos13091421/s1, Table S1: Linear Regression Equation and T-test Analysis result for the Bay of Bengal and the Arabian Sea to obtain the Combined Intensity Dataset; Figure S1: Geopotential height and wind circulation for the Pre-monsoon (Top row), Monsoon (Middle row) and Post-monsoon (bottom row) seasons at 200 hPa (left column); 500 hPa (middle column) and 1000 hPa (right column); Figure S2: Difference between the EARLY (1989–2003) and LATE (2004–2018) periods for (a) All Months; (b) Pre-Monsoon period (February–May); (c) Monsoon period (June–September); and (d) Post-Monsoon period (October–January). Density Maps are calculated at 0.1° resolution; Figure S3: (a) EARLY (1999–2008); (b) LATE (2009–2018), and (c) Difference between the EARLY and LATE periods for the frequency of at least 34 kt winds after TC landfall. Density Maps are calculated at 0.1° resolution; Figure S4: Extent of over land rainfall (mm) for the two 10-year periods: (a) 1999 to 2008; (b) 2009 to 2018; and (c) their difference; and Figure S5: Extent of over land TC rainfall (mm) by monsoon period for the 20 years from 1999 to 2018 for (a) the pre-monsoon; (b) main monsoon period; and (c) post-monsoon.

**Author Contributions:** R.K. is the primary author who processed the data. R.K. also mapped, analyzed the results, and drafted the manuscript. E.A.R. contributed to the analysis and discussion. and provided constructive guidance on the research and editorial input for this article. C.S. provided the DAV wind dataset and contributed to software development. All authors have read and agreed to the published version of the manuscript.

**Funding:** This research received no external funding.

**Institutional Review Board Statement:** Not applicable.

**Informed Consent Statement:** Not applicable.

**Data Availability Statement:** The original contributions presented in the study are included in the article/supplementary material; further inquiries can be directed to the corresponding authors.

**Acknowledgments:** The first author is supported by a UNSW Postgraduate Research Scholarship. Zlatko Jovanowski provided guidance on the development of the regression method used to combine the JTWC and IMD best track datasets into one dataset for use in this paper. This research is undertaken with the assistance of resources and services from the National Computational Infrastructure (NCI), which is supported by the Australian Government. The IBTrACS data were made available by NOAA/NCDC, and the TRMM rainfall dataset was provided by NASA.

**Conflicts of Interest:** The authors declare no conflict of interest.

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
