# Peer review of "Tropical Cyclone Exposure in the North Indian Ocean"

_atmosphere, doi:10.3390/atmos13091421_

Round 1

Reviewer 1 Report

This manuscript analyzed the changing impact of TCs in the NIO by a new satellite-based data set. The results reveal some interesting phenomena and the analysis process is reasonable, but the English writing should be improved and the analysis associated with the IOD should be discussed more. Following are the comments.

Comments:

1.      The authors introduce that both ENSO and IOD can influence the geneses and development of the TCs in the NIO, but the authors just analysis the influence of IOD in the manuscript. More discussions about the influence of ENSO and IOD and the differences should be added in the manuscript to help the readers to understand.

2.      The study period about the TCs are divided into the pre-monsoon (March-May) , monsoon (June-September) , and post-monsoon (October-January) in figure 4, But I suggest the authors to analysis it for three periods with the earlier-monsoon(May-June) and main-monsoon(July-September), and post-monsoon (October-January) following figure 3b and the previous studies mentioned in the introduction and summary.

3.      The citation style in the manuscript should be following the criteria of the journal ie. Line 74, line 86, line 241 ,line 552, line 558 and others.  

4.      Line 17 “TC activity is relatively lower compared to the pre-and post-monsoon periods” is not consistent with that in figure 3b.

5.      Line 37, what does “high seas” mean?

6.      Lines 59-61, do you mean “On average approximately 5.5 TCs develop in the NIO annually with four in the Bay of Bengal and one and a half in the Arabian Sea [18-19], which comprises about 7% of the global annual total TCs”?

7.      Lines 83-90, the sentences are too long and the meaning are not clear.

8.      Lines 93-97, not clear.

9.      Line 111, associated withshould be “due to”

10.   Line 164, “water” should be “ocean”.

11.   Line 379-382, not clear.

Author Response

Response to Reviewer 1 Comments

Comments

Response

Manuscript ID: atmosphere-1873758
Title:   Tropical Cyclone exposure in the North Indian Ocean

Point 1: The authors introduce that both ENSO and IOD can influence the geneses and development of the TCs in the NIO, but the authors just analysis the influence of IOD in the manuscript. More discussions about the influence of ENSO and IOD and the differences should be added in the manuscript to help the readers to understand.

Response: Thank you so much for the comment. The more detailed discussion has been added in lines 79-85 and 91-93 of the revised version of the manuscript.

Point 2: The study period about the TCs are divided into the pre-monsoon (March-May) , monsoon (June-September) , and post-monsoon (October-January) in figure 4, But I suggest the authors to analysis it for three periods with the earlier-monsoon(May-June) and main-monsoon(July-September), and post-monsoon (October-January) following figure 3b and the previous studies mentioned in the introduction and summary.

Response: Thank you. The reviewer raises an interesting point. The peak monsoon period in India is defined variously as starting 1 June, 15 June, and 1 July depending on where the location is (further inland for later starting dates). In this study, we have been careful to follow the periods as defined in [16], which define the periods based on the onset of the monsoon itself.  We then look at how TC activity changes in relation to the changing large-scale circulation during these periods.   We are confident that this is the right way to do this analysis rather than defining periods based on TC activity and so we have decided to keep our original defined periods.  The one exception is that based on a comment from Reviewer 2, we have added “February” into the pre-monsoon period.  Since no TCs made landfall during this month in the 30 years of our study, the actual analysis doesn’t change.

Point 3: The citation style in the manuscript should be following the criteria of the journal ie. Line 74, line 86, line 241 ,line 552, line 558 and others. 

Response: Thank you for finding these. In the revised manuscript, these lines have been corrected by following the MDPI atmosphere journal criteria.

Point 4:  Line 17 “TC activity is relatively lower compared to the pre-and post-monsoon periods” is not consistent with that in figure 3b.

Response:  Thanks to the reviewer for pointing this out. We have altered line 17 to read “… TC activity is relatively lower compared to the post-monsoon period …”

Point 5:  Line 37, what does “high seas” mean?

Response: Thank you for the question. In Line 38, “High seas” is commonly used to refer to large wind-driven waves that can cause the sea state to become very dangerous for maritime activities.

Point 6:   Lines 59-61, do you mean “On average approximately 5.5 TCs develop in the NIO annually with four in the Bay of Bengal and one and a half in the Arabian Sea [18-19], which comprises about 7% of the global annual total TCs”?

Response: Yes, this is what is intended.  We have now modified lines 60-61 in the manuscript as suggested by the respected reviewer.

Point 7:  Lines 83-90, the sentences are too long, and the meaning are not clear.

Response: Thank you. We now changed the lines to make the sentences simple and meaningful to the reviewer in the revised version of the manuscript in lines 93-106 to read “However, TC genesis is….. in Negative IOD phases [32].”

Point 8: Lines 93-97, not clear.

Response: Thanks again. The sentence has been shortened into two sentences and re-arranged to make the meaning clear in lines 109-111 to read “This is, in part,…..TC period.” 

Point 9: Line 111, “associated with” should be “due to”

Response: In the revised version, the correction has been made in line 125.

Point 10: Line 164, “water” should be “ocean”.

Response: Line 178 in the revised manuscript has now been fixed based on the comment.

Point 11: Line 379-382, not clear.

Response: Thank you. The lines 396-399 in the revised manuscript have been revised to clarify that there is high variability in annual landfalls that occur during negative IOD years.

Reviewer 2 Report

Review of “Tropical Cyclone Exposure in the North Indian Ocean,” Kabir et al.

Recommendation: Minor Revisions.

This article focuses on tropical cyclone exposure (wind and rainfall impact) in the North Indian Ocean basin. The region is vulnerable to tropical cyclones due to its high coastal population and low-lying coastlines with multiple river deltas. The study deals with geographical, seasonal, Indian Ocean Dipole (IOD), and climate trend variability.

The study is well conceived and executed, and the article is well written and comprehensive. In particular, using the DAV method to estimate the wind radii and wind exposure is a novel approach (though a bit uncertain for individual storms, see #5 below).

There are a few minor considerations for the authors to address, mainly with figure legibility and textual clarifications:

1. The font size on several figures is a bit small, especially the legend and color bar labeling. Specifically, I had trouble reading parts of Figs. 2, 4, 5, 9, 10, and 11.

2. The markers for tropical cyclone genesis locations in Figs. 4 and 7 should be larger.

3. Some figure panel labels obstruct the underlying patterns. It would be good to place figure panel labels above the panels rather than inside them.

4. The methods used for identifying tropical depressions in the North Indian basin are not the same as used by the National Hurricane Center for the North Atlantic and eastern North Pacific basins, especially the IMD data. Some of the tropical depressions may be monsoon depressions that would not be tracked as TDs by the NHC. This is not a major flaw, but it should be pointed out somewhere in the article.

5. While the DAV method is a novel approach for statistics and trends involving many storms, it has greater uncertainties for individual storms and may not be suitable as guidance to stakeholders for individual storms. This should be pointed out in Sec. 2.6.

6. Line 54: February was curiously left out. It is considered pre-monsoon, post-monsoon, or a transition month?

7. Line 214: Does “begin developing” here mean the positive or negative phase?

8. Line 522: “not statistically significant?”

Author Response

Response to Reviewer 2 Comments

Comments

Response

Manuscript ID: atmosphere-1873758
Title:   Tropical Cyclone exposure in the North Indian Ocean

Point 1:  The font size on several figures is a bit small, especially the legend and color bar labeling. Specifically, I had trouble reading parts of Figs. 2, 4, 5, 9, 10, and 11.

Response: Thank you for your comment. In the revised manuscript, Figures 1(a), 1(b), 2(b), 4(b), 5, 9, 10, and 11 have been replotted with clear legends and labeling.

Point 2:  The markers for tropical cyclone genesis locations in Figs. 4 and 7 should be larger.

Response: Thank you. In the revised manuscript, panels (b) in Figure 4 and Figure 7 have been revised to make the genesis locations clearer.

Point 3:  Some figure panel labels obstruct the underlying patterns. It would be good to place figure panel labels above the panels rather than inside them.

Response: Thanks. The panel labels, which were obscuring information in plots have been moved for Figures 1, 4, 7, 9, 10, and 11.

Point 4:   The methods used for identifying tropical depressions in the North Indian basin are not the same as used by the National Hurricane Center for the North Atlantic and eastern North Pacific basins, especially the IMD data. Some of the tropical depressions may be monsoon depressions that would not be tracked as TDs by the NHC. This is not a major flaw, but it should be pointed out somewhere in the article.

Response: Thank you very much for the comment. We are not sure about our respected reviewer’s thoughts on this point.  We are using combined data from IMD and the JTWC, not from NHC. Differences in methods and definitions among Operational Centers are not the topic of this paper. Therefore, we think any reference to the NHC method of tracking in comparison to that of IMD or JTWC would be out of place.

Point 5:   While the DAV method is a novel approach for statistics and trends involving many storms, it has greater uncertainties for individual storms and may not be suitable as guidance to stakeholders for individual storms. This should be pointed out in Sec. 2.6.

Response: Thank you for the comment. The DAV has shown itself to be quite accurate for individual storms (Dolling et al. 2016; Stark et al. 2019; 2022). However, similar to techniques for estimating intensity or other parameters in use by Operational Centres, errors can be higher for some storms as well, which is why mean errors end up being a smoothing of the two. The DAV is not being used to estimate individual storms in this paper.  But if it was, the DAV would be an acceptable method to estimate the extent of 34-kt winds in the absence of direct observations.

Point 6:    Line 54: February was curiously left out. It is considered pre-monsoon, post-monsoon, or a transition month?

Response: We would like to thank the reviewer for pointing this out.  Because no TCs made landfall in February during the 30-year period, we hadn’t included it in the labeling.  However, this oversight is now rectified in line 54 and line 333 by including February as part of the “pre-monsoon period” here and elsewhere in the manuscript where appropriate.

Point 7:   Line 214: Does “begin developing” here mean the positive or negative phase?

Response: Thank you. Yes, we meant the positive phase. This has now been revised in the manuscript by adding ‘the positive IOD phase begins developing……’ in line 228.

Point 8:  Line 522: “not statistically significant?”

Response: Thank you again. So that there is no confusion with a “statistical test of significance” we have modified the sentence to say “… the trend is weak” in line 543.

Round 2

Reviewer 1 Report

The authors have revised all of the comments in my first time of review. I belief the manuscript can be accepted for publication in Atmosphere。